# Development of an Improved YOLOv7-Based Model for Detecting Defects on Strip Steel Surfaces

**Rijun Wang** [1,2,*], **Fulong Liang** [1,2], **Xiangwei Mou** [1,2,*], **Lintao Chen** [1,2], **Xinye Yu** [1,2], **Zhujing Peng** [1] **and Hongyang Chen** [1]

1    Teachers College for Vocational and Technical Education, Guangxi Normal University, Guilin 541004, China
2    Key Laboratory of AI and Information Processing, Hechi University, Hechi 546300, China
*    Correspondence: rijunwang@mailbox.gxnu.edu.cn (R.W.); xwmou@mailbox.gxnu.edu.cn (X.M.)

**Abstract:** The detection of defects on the surface is of great importance for both the production and the application of strip steel. In order to detect the defects accurately, an improved YOLOv7-based model for detecting strip steel surface defects is developed. To enhances the ability of the model to extract features and identify small features, the ConvNeXt module is introduced to the backbone network structure, and the attention mechanism is embedded in the pooling module. To reduce the size and improves the inference speed of the model, an improved C3 module was used to replace the ELAN module in the head. The experimental results show that, compared with the original models, the mAP of the proposed model reached 82.9% and improved by 6.6%. The proposed model can satisfy the need for accurate detection and identification of strip steel surface defects.

**Keywords:** defect detection; YOLOv7; deep learning; ConvNeXt; attention pooling module





## 1. Introduction

As an important raw material of industry, strip steel is widely used in the production of machinery, aerospace, automotive, defense, light industry, etc. [1]. However, limited by the quality of raw materials, production environment, equipment, manual errors, etc., the strip steel can lead to a variety of problems, the most common one being surface defects [2–4]. The surface defects are an important indicator for manufacturers and customers or consumers to judge the quality of strip steel. In general, the surface defects of strip steel, including crazing, inclusion, patches, pitted surface, rolled-in scale, scratches, and these defects have an impact on the aesthetics of the steel, but more importantly, they reduce the strength, toughness, corrosion resistance and wear resistance of the strip steel [5–7]. In addition, the defects will also affect the strip steel sales of enterprises and may even bring personal safety risks to users [8]. Therefore, the detection and identification of strip steel defects have become a hot issue for scholars to study.

The traditional defect detection methods for strip steel surfaces include manual inspection methods, non-destructive testing methods [9], and frequency flash detection methods [10]. Manual inspection requires inspectors to identify a large number of strip steel defects through the naked eye, which needs a lot of labor due to the complex diversity of defects. Secondly, the large amount of repetitive work makes the inspectors prone to visual fatigue, which can lead to missed inspections and false inspections [11]. Due to the traditional methods existing in low efficiency, error, high requirements for the skills of the inspector, and other shortcomings occur. In recent years, based on deep learning, image processing, target detection, and other automated technologies have begun to gradually replace traditional methods. Deep learning-based target detection can obtain higher recognition accuracy and detection speed, which greatly improves the efficiency of defect detection in real factories. Among them, the You Only Look Once (YOLO) algorithm series has become a popular method in the current target detection research field because of its ability to maintain good detection accuracy despite its fast detection speed.

The causes of defects on the strip steel surface are numerous, and the morphology of defects is complicated. According to the characteristics of the defect shapes, the defects can be roughly divided into three categories: point, line, and surface. Typical defects (shown in Figure 1) can be summarized as crazing, inclusion, patches, pitted surface, rolled-in scale, and scratches. The crazing (as shown in Figure 1a) is caused by excessive surface burning, decarburization, loosening, deformation, and a high content of sulfur and phosphorus impurities on the surface during processing. The crazing generally appears as water ripples or fish scale, which is different from the cracks caused by loose oxide skin. The inclusion (as shown in Figure 1b) is usually caused by the presence of inclusions (metallic or non-metallic) during the strip steel rolling. This defect occurs when the inclusions are fractured or exposed. The size of inclusion defects is related to the number of inclusions, and the edges are relatively clear, usually gray-white, yellow, or brown. The formation of patches (as shown in Figure 1c) is related to the incomplete cleaning of iron oxide on the surface of strip steel and also related to the failure to remove the residual liquid in the annealing process in time. The patches are usually black with large and different shapes. The distribution of patches on the surface of strip steel is random. After the strip steel has been rolled, the iron oxide comes off its surface, resulting in a continuous rough surface called a pitted surface (as shown in Figure 1d). The pitted surface usually appears as dents of different sizes and depths, with dotted distribution or periodic distribution. There are two main reasons for the formation of the rolled-in scale (as shown in Figure 1e); one is that foreign materials are on the surface of the roller, which makes the surface of the strip steel bulge during the roll-forming process. The other one is the low hardness of the roller, which causes the strip steel surface material to come off during the roll forming process; the strip steel surface appears depressed. The scratches (as shown in Figure 1f) is due to the action of external forces or scratches by sharp objects during transportation.

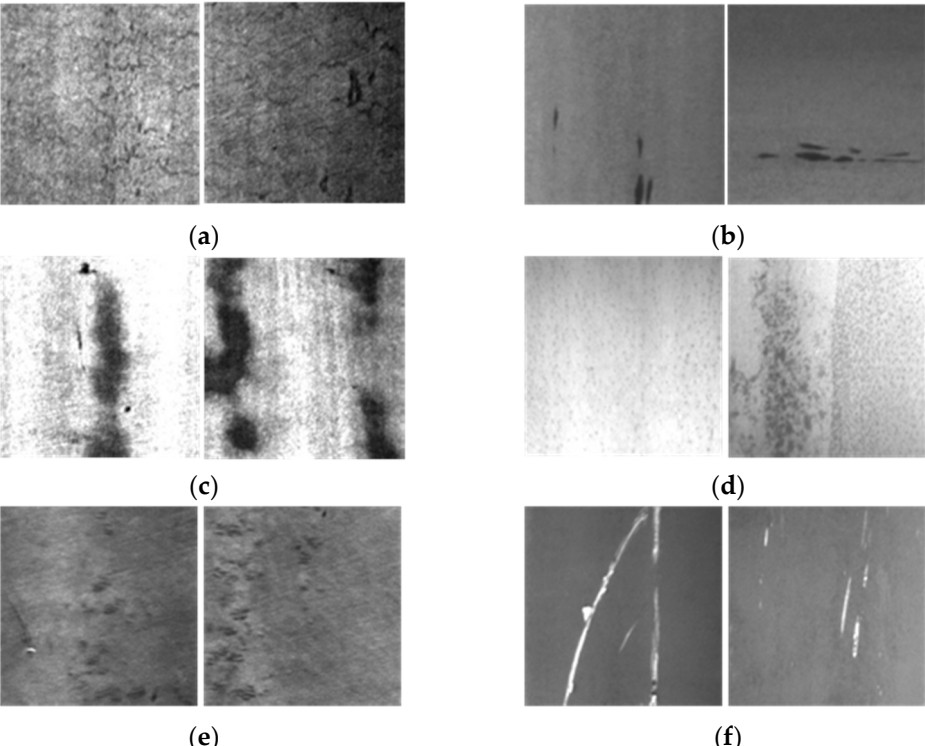

**Figure 1.** Typical defects of the strip steel. (**a**) crazing; (**b**) inclusion; (**c**) patches; (**d**) pitted surface; (**e**) rolled-in scale; (**f**) scratches.

All of the above defects have a negative impact on the integrity and functionality of the strip steel. As we can see in Figure 1, these defects present a variety of types. Some of the defects are small and vary in size. The same type of defect presents different

characteristics due to different causes. Moreover, the distinction between defects is not clear enough. Therefore, it is extremely difficult to identify defects on the surface of the strip steel accurately.

Identifying defects on the strip steel surface is an important criterion for judging the quality of the strip steel and also facilitates the producer in finding the source of the problem for further improvement. It is of great importance for the production and manufacturing of strip steel. However, traditional methods are no longer applicable in today's progressively intelligent era, and the rapid development of computers has brought a great impetus to the field of computer vision. A fast regularity metric for defect detection in non-textured and uniformly textured surfaces is proposed by Tasi et al. [12]. This method is used to detect defects only through a single discriminant feature. It avoids the use of complex classifiers in a high-dimensional feature space. On the other hand, the method does not require learning from a set of defective and non-defective training samples. Liu et al. [13] proposed a new Haar–Weibull variance (HWV) model for unsupervised steel surface defect detection. The anisotropic diffusion model is used to eliminate the influence of patches, and then a new HWV model is developed to characterize the texture distribution of each local patch in the image, thus forming a parametric distribution to extract the background in the image effectively. In order to solve the under-segmentation or over-segmentation problem, a global adaptive percentile threshold method for gradient image is proposed in the literature [14]. Without considering the defect size, this method can adaptively change the percentile used for thresholding and retain the characteristics of defects. A defect detection model using an optimal Gabor filter was proposed by Tong et al. [15]. By using an optimal Gabor filter, the model can significantly reduce the computational cost and operate in real time to solve the problem of fabric detection. Choi et al. [16] applied the Gabor filter to the detection of porous defects in steel plates, and the classification performance of defects was improved with the use of the double threshold method. An entity sparsity tracking (ESP) method for identifying surface defects is proposed by Wang et al. [17] capable of detecting surface defects in an unsupervised manner.

In order to accelerate industrial production, improve product quality and save labor, many researchers have devoted themselves to applying deep learning target recognition methods to industrial production. Current deep learning-based target detection algorithms are basically divided into two categories, namely, the one-stage target detection algorithms and the two-stage detection algorithms. The two-stage target detection algorithm can be roughly divided into two steps. The first step is to locate the target in a candidate frame, and the second step is to make a final prediction of the target. The two-stage target algorithm includes the regional convolutional neural network (R-CNN) [18], Fast R-CNN [19], and Faster R-CNN [20]. Compared to the two-stage target detection algorithm, the one-stage target detection algorithms directly predict the location and category of the target, which is simpler and more direct. The one-stage target algorithm includes single shot multiBox detector (SSD) [21], RetinaNet [22], and YOLO series algorithms [23–30]. In order to meet the requirements of strip steel surface defect detection, an improved model [31] is proposed by combining the improved ResNet50 [32] with Faster R-CNN. The experimental results showed that the accuracy of detection was as high as 98.2%. However, the proposed model has a large number of parameters, which makes the algorithm inference runtime longer. A model of YOLOV4 based on an attention mechanism is proposed by Li et al. [33]. The model has a stronger feature extraction capability; the average accuracy reached 85.41% in detecting four types of strip steel defects. The TRANS module based on Transformer [34] was added to the backbone and detection head of the YOLOv5 model, and an improved Transformer-based YOLOv5 model was proposed by Guo et al. [35]. The test results showed that the average detection accuracy is 75.2%, improving about 18% compared to Faster R-CNN.

In summary, the target detection algorithm based on deep learning can effectively solve the problem of strip defect detection. Based on the above work, an improved YOLOv7-based model for detecting defects on strip steel surfaces is proposed in this paper. By

introducing the ConvNeXt module to the backbone network, the ability of the network to extract defect features and accelerate network inference is enhanced. By embedding the CBAM into the MP layer of the model detection head, an attention-pooling structure is formed to enhance the ability to cope with complex and different strip steel surface defects.

The organization of this paper includes the following sections. In Section 1, the significance of the research and the contribution of this paper is given. A detailed summary of the research results related to the field of strip steel defect detection is analyzed, especially based on deep learning. A detailed description of the original You Only Look Once version 7 (YOLOv7) model, the loss function, and label assignment is given in Section 2. In Section 3, an improved model based on YOLOv7 was developed for detecting defects on the strip steel surface is described in detail. In Section 4, the detailed experimental results are described. The conclusion is given in Section 5.

Our contributions are as follows:

1. An improved YOLOv7-based model for detecting defects on strip steel surfaces is proposed.
2. To enhance the network's ability to extract defects features and speed up network inference, the ConvNeXt module is introduced to the backbone network of the YOLOv7 model.
3. To reduce the amount of operations and simplify the network structure, the Efficient Layer Aggregation Network (ELAN) module in the detection head of the YOLOv7 model is replaced by an improved C3 module (C3C2).
4. By embedding the Convolutional Block Attention Module (CBAM) into the maximum pooling (MP) layer of the model detection head, an attention pooling structure is formed to enhance the ability to cope with complex and different strip steel surface defects.

## 2. Methodology

### 2.1. YOLOv7 Network Structure

The YOLOv7 [30] (version 0.1) network structure is based on YOLOv5 [27] (version 5.0), which introduces the idea of model re-parameterization. In YOLOv7 network structure, deep supervision technique is added, dynamic label assignment strategy is improved, coarse-to-fine guiding label assignment strategy is proposed, etc. Among them, the role of the model re-parameterization is splitting a whole module into several identical or different module branches during the training process and integrating several branch modules into a fully equivalent module during the inference process. The benefit of model re-parameterization is that better feature representations are obtained, computational and parametric quantities are reduced, and inference speed is improved.

Deep supervision is a common technique used in deep network training. The main idea of deep supervision is to add an auxiliary head in the middle layer of the network. The shallow network weights and auxiliary losses are used as a guide to supervising the backbone network. Thus, the problems of disappearing training gradients and slow convergence of deep neural networks are solved (the YOLOv7 model explored in this paper does not have an auxiliary training head).

The coarse-to-fine guiding label assignment strategy is used to make the label assignment more accurate. The strategy is guided by the prediction results of the lead head to generate coarse-to-fine hierarchical labels. The coarse-to-fine hierarchical labels are used for auxiliary head and lead head learning, respectively.

The overall network structure of YOLOv7 (as shown in Figure 2) is very similar to YOLOv5; the main difference between them is the internal components of the network. Firstly, in the backbone part, the extended efficient layer aggregation network (E-ELAN) and MP structure are used in the backbone part of the network. Secondly, the neck layer and the head layer are merged, still called the head layer. The YOLOv7 network extracts image features mainly through the backbone part of the E-ELAN and MP structure. The authors of the original paper believe that the deeper the network is, the better it is for network

learning and convergence. A more efficient network can be built by controlling the shortest and longest gradient paths in the network. Thus, after comparing with VoVNett [36], CSPVoVNet [37], and ELAN [38], the E-ELAN (an extended version based on ELAN) is proposed. The E-ELAN only changes the structure of the computational module, while the structure of the transition layer is completely unchanged. By using the strategy of expand, shuffle, and merge cardinality, the network learning capability is continuously enhanced without destroying the original gradient path. Unlike the previous network structure of YOLO, the MP layer in the YOLOv7 network structure uses both maxpooling and $3 \times 3$ convolution with stide = 2 to downsampling. The outputs are concatenated by means of concat, which allows the network to extract features better.

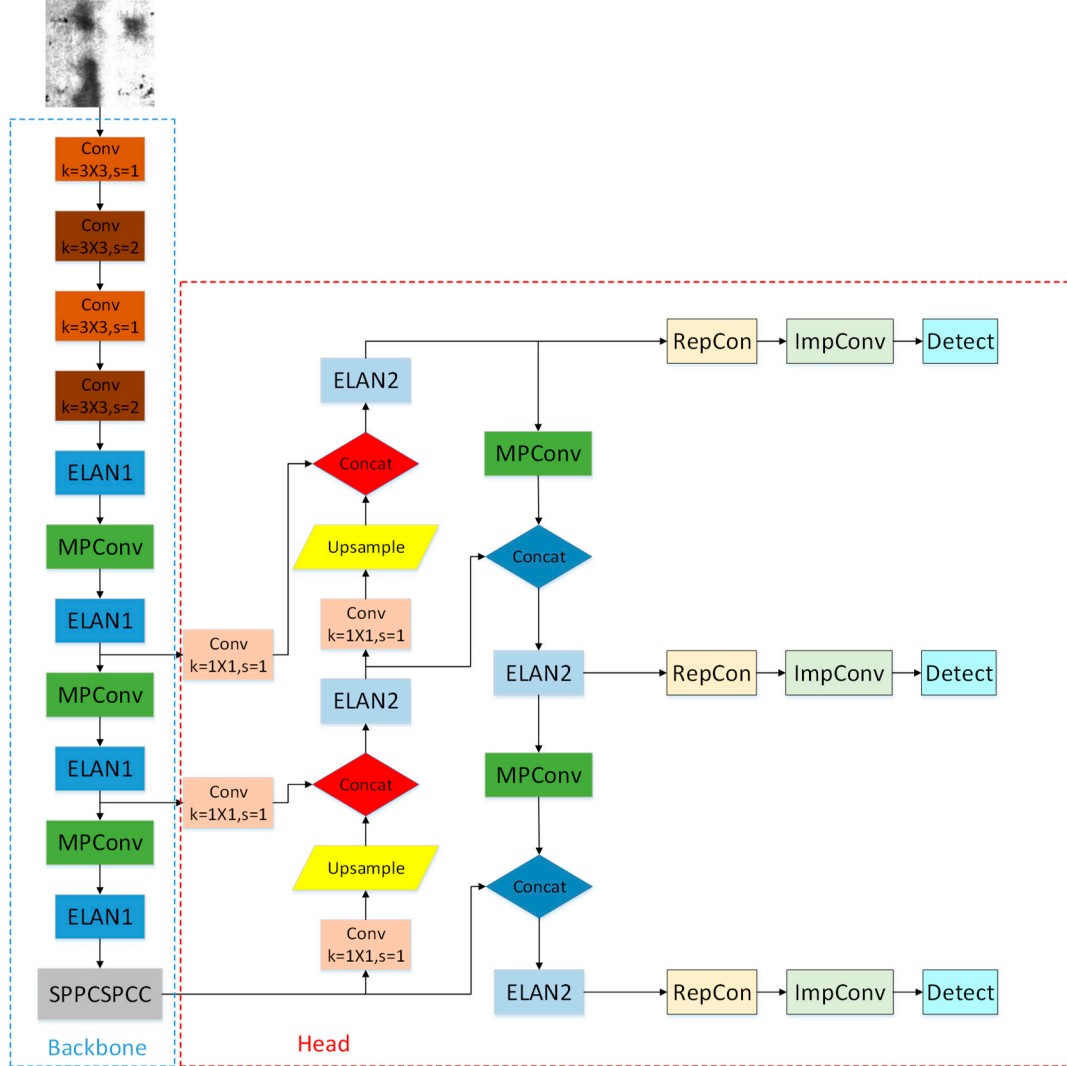

**Figure 2.** YOLOv7 network structure.

The procedure of strip defect detection with YOLOv7 is as follows:

1. Using a camera with higher resolution to collect pictures of the strip steel with defects on the surface.
2. Using the labelimg tools to process the defects that appear in the strip steel on these images, frame them accurately with a rectangular box and mark the category.
3. Dividing the processed images into the training set, test set, and validation set according to a certain ratio; putting the training set and validation set into the model of YOLOv7 for training and validation; and using the test set to test the model training effect.

*2.2. Loss Function and Label Assignment*

The overall loss function of YOLOv7 remains the same as YOLOv5. The loss function is divided into three parts: the classification loss $L_{cls}$, the objective confidence loss $L_{obj}$ and the localization loss $L_{loc}$.

The binary cross entropy (BCE) loss is used for classification loss $L_{cls}$, and note that only the classification loss of positive samples is calculated. The objective loss $L_{obj}$ is still BCE loss; note that the *obj* here refers to the complete intersection over union (CIoU) of the target bounding box and GT Box of the network prediction. The objective loss $L_{obj}$ is calculated here for all samples. The localization loss $L_{loc}$ is used as CIoU loss, and note that only the location loss of positive samples is calculated.

Therefore, the loss function of YOLOv7 can be described as follows:

$$Loss = \lambda_1 L_{cls} + \lambda_2 L_{obj} + \lambda_3 L_{loc} \tag{1}$$

where $\lambda_1$, $\lambda_2$, $\lambda_3$ are the equilibrium coefficients.

A new method of label assignment is used in the YOLOv7 network structure. Using the prediction of the lead head as a guide, coarse-to-fine hierarchical labels are generated. The labels are used for the learning of the auxiliary head and the lead head, respectively. The lead head has a stronger learning capability, allowing the auxiliary head to learn the information already learned by the lead head directly, and the lead head can focus more on the residual information that has not yet been learned. The details can be seen in Figure 3a,b.

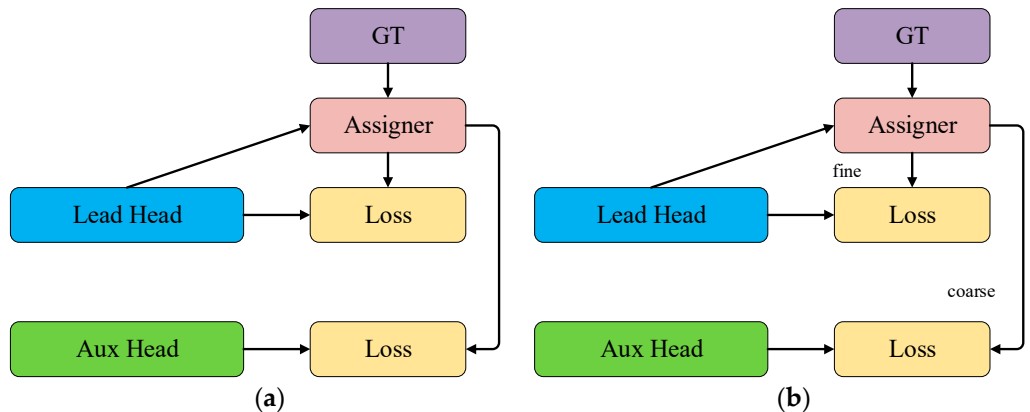

**Figure 3.** New label assignment method. (**a**) Lead guided assigner; (**b**) Coarse-to-fine guided assigner.

## 3. Improvement of YOLOv7

The improved YOLOv7 algorithmic network structure architecture proposed in this study is marked in the red box in Figure 4, and detailed information is given in the subsequent two sections.

The specific improvement points of the YOLOv7 algorithm structure in this study are marked with different colored rectangular boxes in Figure 4. The ConvNeXt module in the purple box is added to the head of YOLOv7 to enhance the ability of the model to extract features, and the C3C2 module in the yellow box replaces the ELAN structure in the original structure of YOLOv7 to streamline the model size, and finally, the green CBAM attention mechanism in the green box is embedded in the first MP layer structure to improve the network's ability to identify minor and inconspicuous defects.

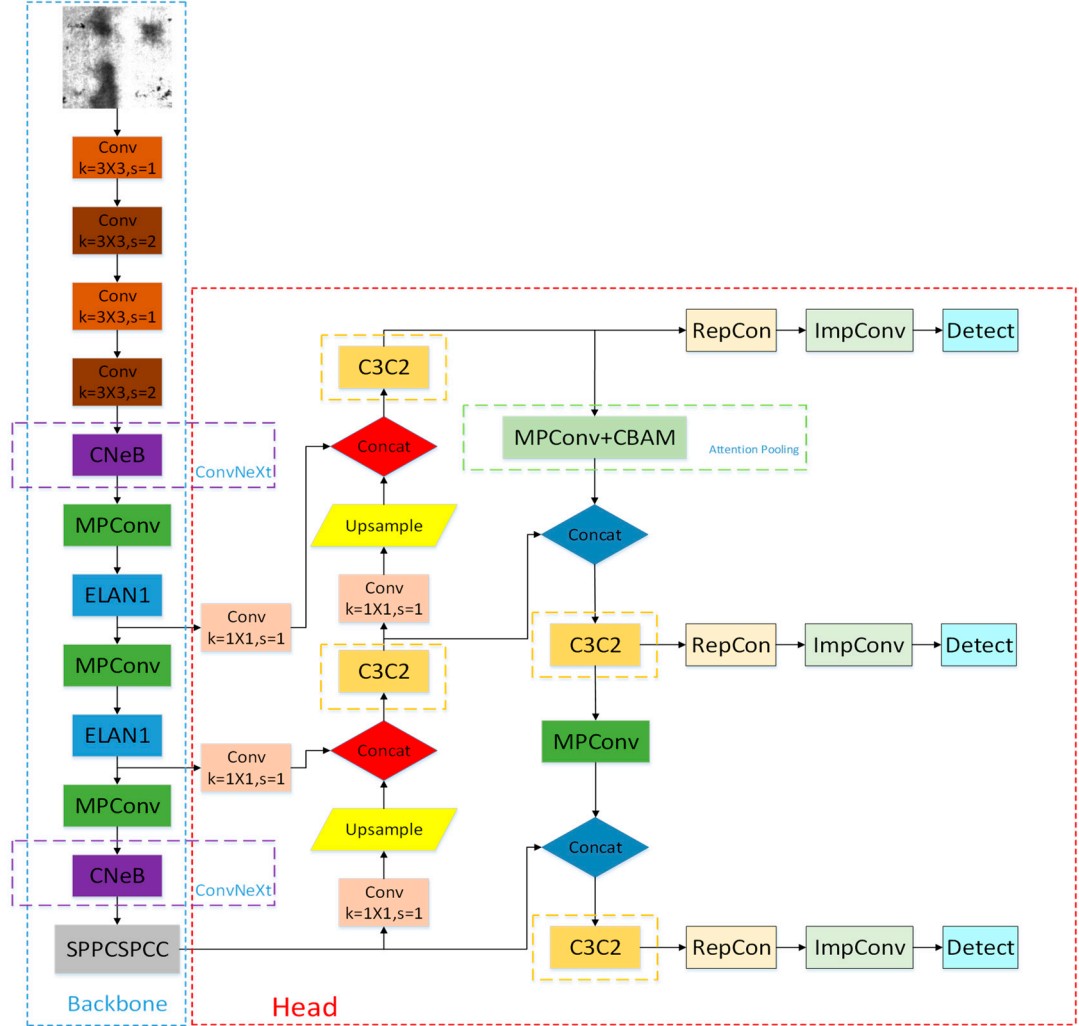

**Figure 4.** Improved YOLOv7 algorithm network structure.

### 3.1. ConvNeXt Module

In order to achieve accurate detection of strip steel surface defects, we applied the network model of YOLOv7 to the strip steel dataset. In order to improve the accuracy of the model, we tried to add the newly proposed ConvNeXt [39] convolutional structure module to the backbone of YOLOv7 for better extraction of strip steel surface defect features. The ConvNeXt has four different versions of T/S/B/L, which are configured as follows:

$$
\begin{aligned}
&\text{ConvNeXt-T: C} = (96, 192, 384, 768), \text{B} = (3, 3, 9, 3) \\
&\text{ConvNeXt-S: C} = (96, 192, 384, 768), \text{B} = (3, 3, 27, 3) \\
&\text{ConvNeXt-B: C} = (128, 256, 512, 1024), \text{B} = (3, 3, 27, 3) \\
&\text{ConvNeXt-L: C} = (192, 384, 768, 1536), \text{B} = (3, 3, 27, 3)
\end{aligned}
\tag{2}
$$

where C represents the number of input channels in the four stages, and B represents the number of repeated stacking blocks per stage.

The computational complexity, structure size, and the number of input channels of the ConvNeXt increase sequentially from version T to version XL. To avoid breaking the entire continuous downsampling structure in the backbone of YOLOv7, after weighing the module size and the number of output channels, we choose to replace the first and last ELAN modules in its backbone with the ConvNeXt-B module. The structure of the ConvNeXt-B module is shown in Figure 5.

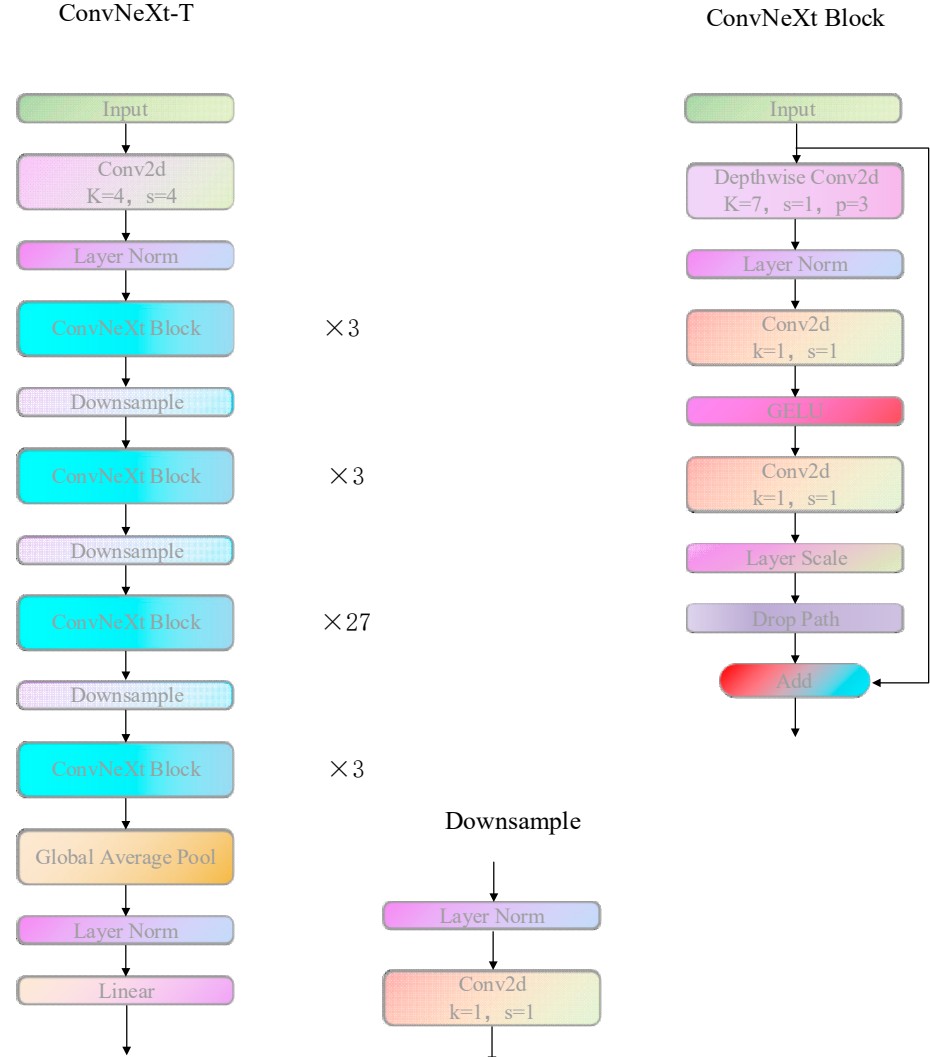

**Figure 5.** ConvNeXt-B module structure.

The network structure of the ConvNeXt module is a pure convolutional network structure based on the ResNet-50 network structure, which is designed based on the structure of the Swin Transformer [40]. While the ConvNeXt module is benchmarked against the Swin Transformer network structure, it also actively learns from the previous classical network structure. For example, the depthwise convolution structure adopted by the ConvNeXt module is learning from the method of ResNeXt [41]. Through the five comparative experiments of macro design, deep residual learning for image recognition (ResNeXt), inverted bottleneck, large kerner size, and various layer-wise micro designs, the network model is gradually optimized. Finally, ConvNeXt is proposed. With the same floating point of operations (FLOPs), the ConvNeXt has faster inference speed and higher accuracy than Swin Transformer.

### 3.2. Improvement of C3(C3C2)

In YOLOv7's head part, the ELAN module is used to extract features from the input feature maps. However, for our strip steel surface defect dataset, there are large differences in the size and shape of defects in the same category. Moreover, there are also similarities between different classes of defects. In addition, due to the different material quality of different strip steel samples and the influence of lighting, the gray value of the intra-class defect image will also change. These reasons make it difficult for the network to extract features. The ELAN module in YOLOv7 is not very effective in extracting the features

of the image. In addition, the ELAN structure contains more convolutional modules and residual connections, which will bring more computation and reduce the inference speed. Therefore, we try to improve the C3 module in the latest version of YOLOv5 (shown in Figure 6a) into the C3C2 module (shown in Figure 6b) and replace the ELAN module in the head part to further enhance the feature extraction and fusion capability of the YOLOv7 network structure. Meanwhile, the parameter computation is reduced, and the inference speed is improved.

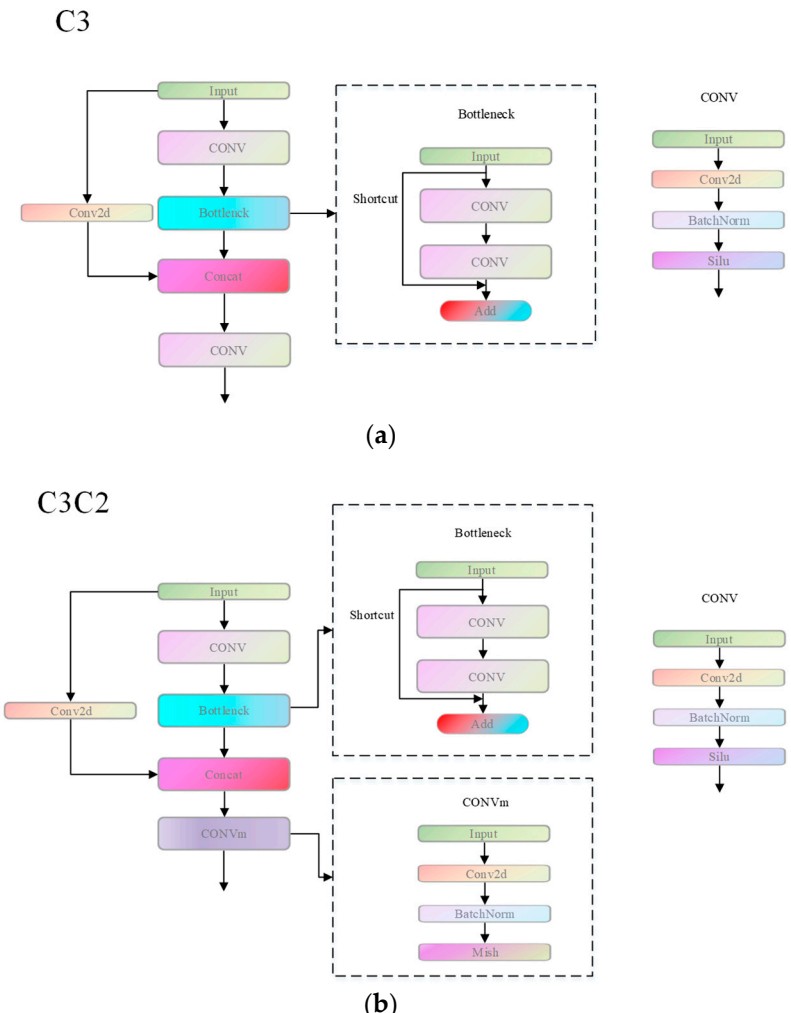

**Figure 6.** Improvement of C3(C3C2). (**a**) C3 module; (**b**) C3C2 module.

The C3C2 convolution module is inspired by Swin Transformer's network structure. It is based on the original C3 module, the residual branch convolution module is changed to a simple convolution structure, and the batch normalization (BN) layer and activation function layer are removed to reduce the amount of parameter calculation. Given that the Mish activation function [42] has a better ability to suppress overfitting than the Silu activation function, the Mish activation function is more robust to different hyperparameters. In view of the above advantages, we changed the activation function in the final convolution module from the original Silu (swish) to the Mish activation function. As a result, the nonlinear variation of the network is enhanced when the convolution module is input after the final Concat operation.

### 3.3. Attention Pooling Module

The MP layer structure, as shown in Figure 7a, is used for downsampling in the YOLOv7 head. The feature map will be downsampled in two branches after entering the

MP layer structure, one branch for the Maxpooling layer with a large convolutional kernel of 2 and the other branch for the convolutional kernel of size 3 with a step size of 2 for downsampling. Finally, the output of the two branches will be Concat operation and then output. Since the strip steel defect dataset contains many small and dense defects, which are not easy to identify, we try to add the attention mechanism CBAM [43] to the MP layer structure to build the attention-pooling module, as shown in Figure 7b. As a result, the network is enabled to focus on more important targets by itself and strengthen the ability of the network structure to identify defects.

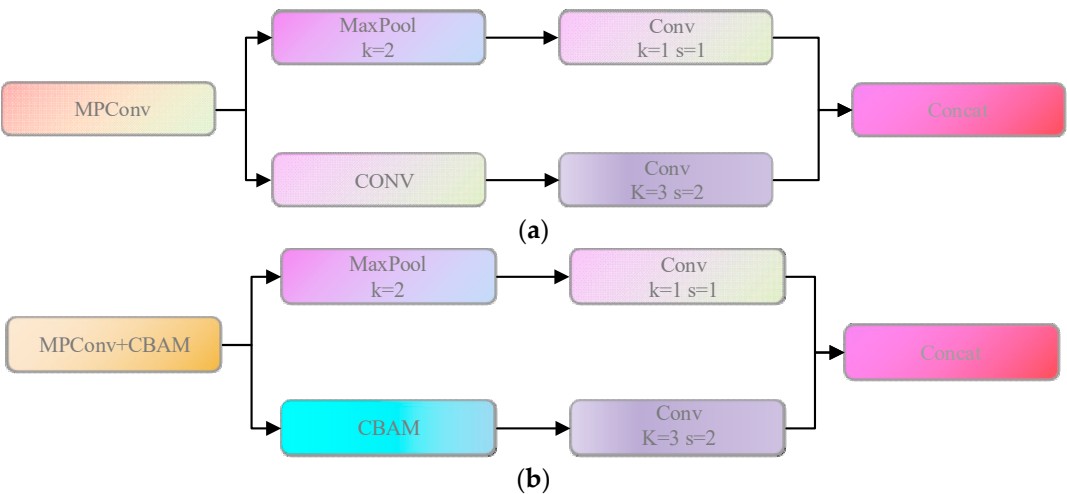

**Figure 7.** Attention pooling module. (**a**) MP layer structure; (**b**) MP layer structure with CBAM.

The CBAM is an attention mechanism module that combines spatial and channel. It can achieve better results than the squeeze-and-excitation networks (SEnets) [44] attention mechanism, which only focuses on channels. The CBAM contains two independent sub-modules, namely, the channel attention module (CAM) and the spatial attention module (SAM). This not only makes the network more capable of focusing on key information but also achieves a plug-and-play effect by weighting attention in both channel and space, respectively.

When the feature map is input to the CBAM, it will first pass through the CAM. In the CAM, the feature map will first pass through two parallel MaxPool and AvgPool layers to compress the feature map into two one-dimensional feature vectors. Next, the two one-dimensional feature vectors are fed into a two-layer shared neural network for activation and channel transformation. The result of the transformation is subjected to an Add operation. Finally, the final channel attention feature is generated by a sigmoid activation operation. The channel attention feature map and the input feature map are multiplied elementwise to generate the input features needed by the SAM. After entering the SAM, the channel-based global max pooling and global average pooling operations are performed to compress the channels of the feature map. Next, the channel-based Concat operation on these two results is carried out. After that, the convolution operation is performed to reduce the dimensionality to one channel. Then, the spatial attention feature map is generated by the sigmoid activation function. Finally, the spatial attention feature is multiplied by the input feature of this module to obtain the final generated feature.

## 4. Experiment and Result Analysis

In this section, the dataset, evaluation metrics, comparison objects, and methods are described, and the experimental results are analyzed to confirm the usefulness of the improved model.

### 4.1. Experimental Details and Dataset

The experimental environment used a computer with the following configuration: Windows 11 operating system, Intel (R) Core (TM) i5-9300 CPU with 2.40 GHz processor, and NVIDIA GeForce GTX 1650 graphics card. The experimental dataset of the NEU-DET [45] dataset of Northeastern University consists of six types of defect image data, including crazing, inclusion, patches, pitted surface, rolled-in scale, and scratches. There are 300 samples for each type of defect, for a total of 1800 grayscale images. For the defect detection task, bounding box annotations are also provided in the dataset, indicating the class and location of defects in each image. A total of 1800 images, the image size is $200 \times 200$, the training set and the test set are divided according to the ratio of 9:1, and then 10% of the training subset is used as the validation set.

### 4.2. Performance Evaluation

To measure the accuracy of target defect detection, we used two metrics (i.e., average precision (AP) and mean average accuracy (mAP)) as performance evaluation criteria. These are calculated as follows.

$$Precision = \frac{TP}{TP + FP} \tag{3}$$

$$Recall = \frac{TP}{TP + FN} \tag{4}$$

$$AP = \int_{0}^{1} P(R)dR \tag{5}$$

$$mAP = \frac{\sum_{i=1}^{c} AP_i}{c} \tag{6}$$

where the *TP* is a true-positive defect, the *FP* is a false-positive defect, and the *FN* is a false-negative defect. The *P(R)* is the precision–recall curve, the *i* is a defect category, and the *c* is the number of defect categories with a value of six in this experiment.

### 4.3. Ablation Research

For the ablation study, we designed a detailed algorithm based on YOLOv7 as the baseline model, the CovNeXt network structure is added to the backbone, the ELAN structure is replaced by the C3C2 structure in the detection head, the CBAM is embedded in the MP layer structure to construct the attention pooling structure. In order to reasonably judge whether the proposed improvements are of application value for strip steel defect detection, the joint ablation experiments on the NEU-DET dataset is carried out. The results are listed in Table 1, where the YOLOv7 represents the original YOLOv7 model. The YOLOv7–ConvNeXt-B represents the replacement of the first and last ELAN modules in the backbone of YOLOv7 with the ConvNeXt-B module. The YOLOv7–C3C2 represents the replacement of the original ELAN structure in the head of the YOLOv7 model with our improved C3C2 structure. The YOLOv7–CBAM represents the addition of the CBAM attention mechanism in the first pooling layer of the head, and the Ours represents the proposed algorithm.

As can be seen from Table 1: (1) after adding the ConvNeXt structure to YOLOv7, although the overall mAP value is not improved, it makes the whole network structure have better recognition ability for the crazing, patches, rolled-in scale, and scratches. (2) For the application of the C3C2 module, the size of the network model is reduced from 36.5 MB to 30.2 MB without reducing the overall detection accuracy of the network. The number of parameters of the network is reduced greatly, and the inference speed of the network speeds up. (3) The AP values of inclusion, patches, rolled-in scale, and scratches are also improved. Finally, the YOLOv7–CBAM greatly enhances the sensitivity of the network to line defects and significantly improves the detection capability of the network for both

crazing and scratch defects. (4) For our proposed algorithm, the AP values of the four types of defects (crazing, patches, rolled-in scale, and scratches) have been improved more significantly, except for a slight decrease in inclusion defects.

**Table 1.** Ablation experiments results.

| | mAP% | AP% | | | | | |
|---|---|---|---|---|---|---|---|
| | | **Crazing** | **Inclusion** | **Patches** | **Pitted Surface** | **Rolled-In Scale** | **Scratches** |
| YOLOv7 | 76.3 | 48.1 | 76.5 | 94.8 | 99.5 | 67.0 | 72.0 |
| YOLOv7–ConNeXt-B | 76.3 | 54.1 | 62.3 | 96.4 | 95.6 | 67.6 | 82.1 |
| YOLOv7–C3C2 | 75.5 | 35.9 | 76.8 | 99.1 | 93.1 | 71.5 | 76.5 |
| YOLOv7–CBAM | 79.4 | 63.6 | 66.7 | 97.7 | 99.5 | 65.5 | 83.6 |
| Ours | 82.9 | 68.9 | 68.3 | 97.8 | 99.5 | 73.3 | 89.3 |

Note: AP = Average precision; mAP = Mean AP; YOLO = You Only Look Once.

### 4.4. Contrasting Experiment

The Precision–Recall (P–R) curves of the YOLOv5, original YOLOv7, and improved YOLOv7 algorithm models for the detection of the six defects in the NEU-DET dataset are shown in Figure 8.

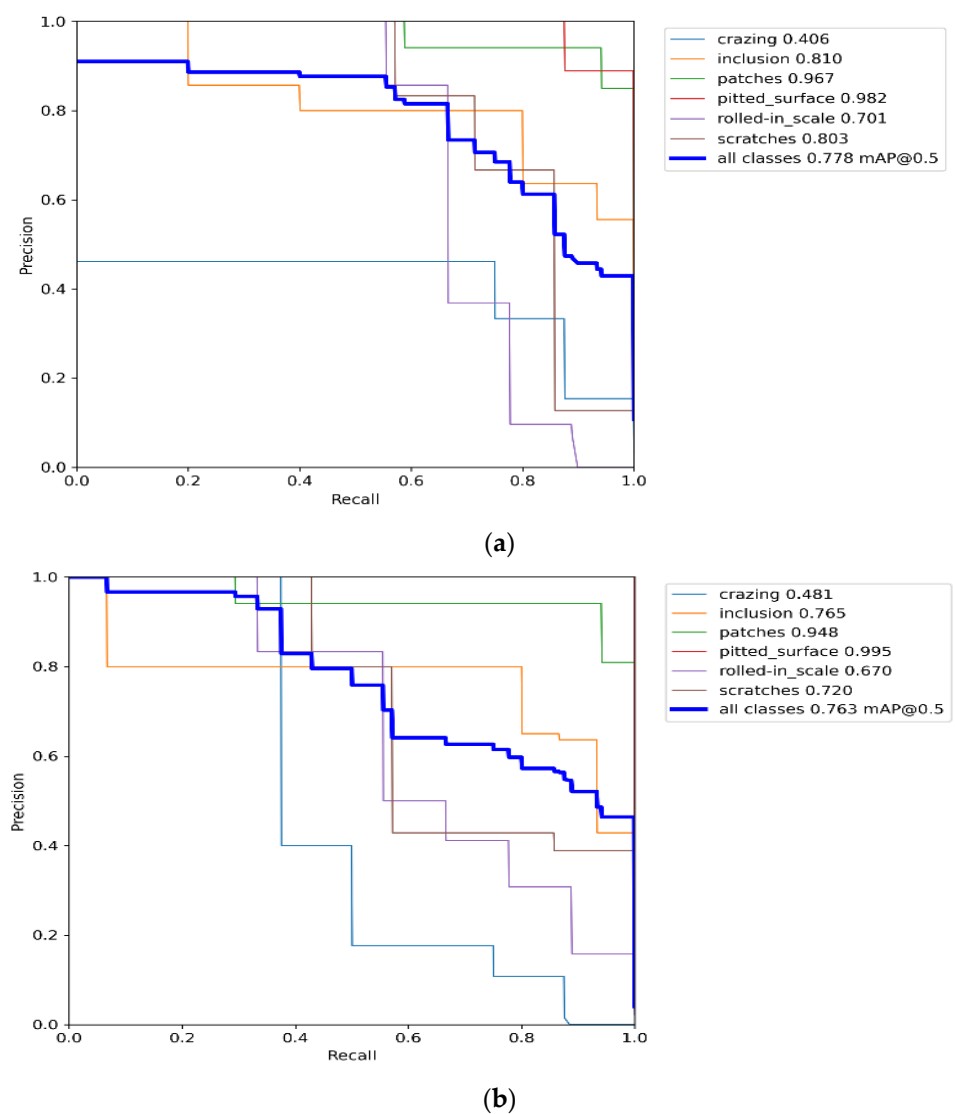

**Figure 8.** *Cont.*

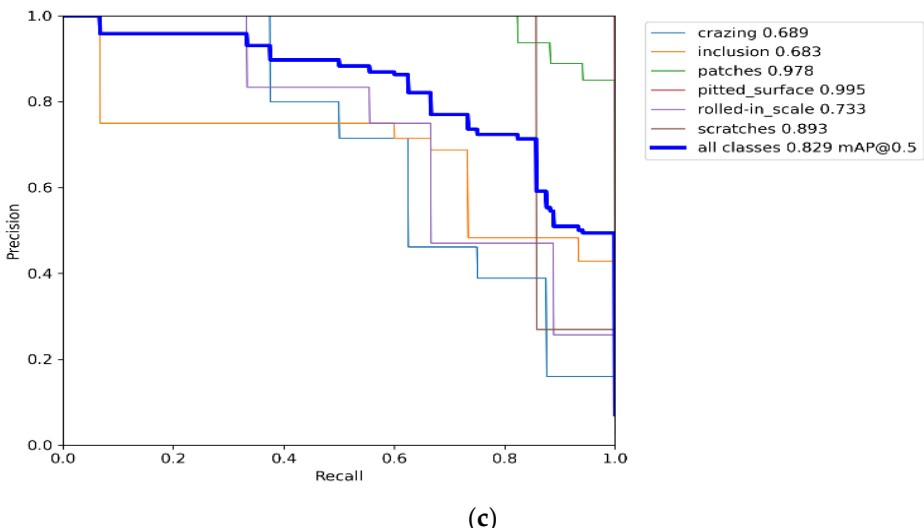

(**c**)

**Figure 8.** Precision–recall (P–R) curves. (**a**) YOLOv5; (**b**) YOLOv7; (**c**) Improved YOLOv7.

The P–R curve is an important indicator to measure the performance of the model. In the P–R curves, the larger the area enclosed by the curve, the better the performance. As can be seen from Figure 8, compared with the other two algorithms, our proposed algorithm has better defect detection performance.

Figure 9 represents a comparison of the visualization results of the detection effects of the above models. It is easy to see that the improved YOLOv7 algorithm model is not only able to locate and find all defects more accurately but also make the prediction frame more precise. Meanwhile, compared with the other two models, the improved YOLOv7 algorithm model is able to improve the false and missed detection very well.

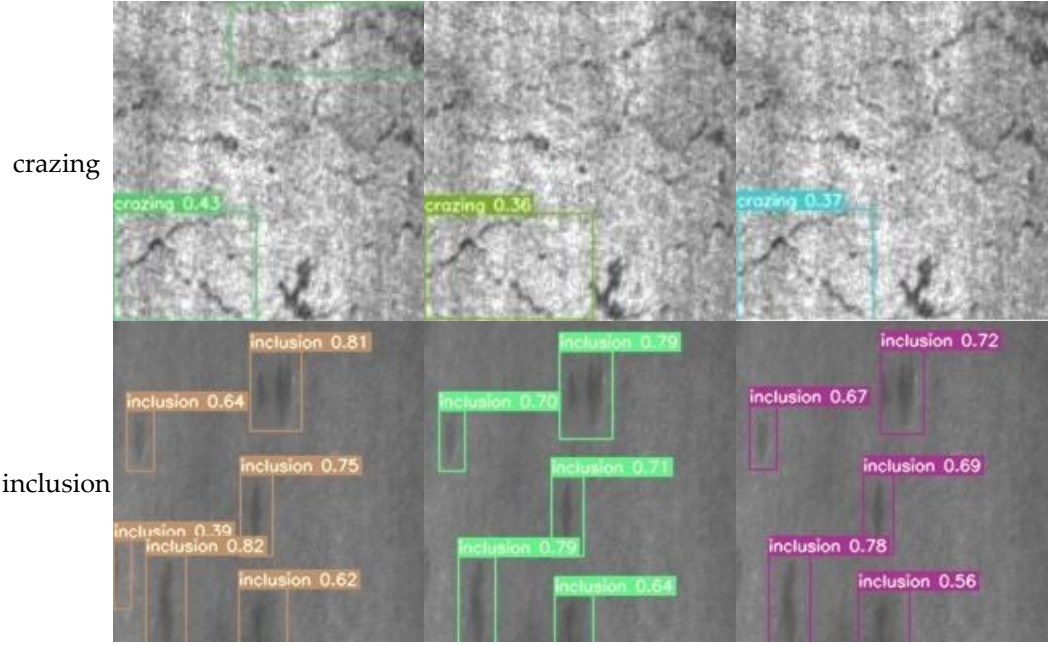

**Figure 9.** *Cont.*

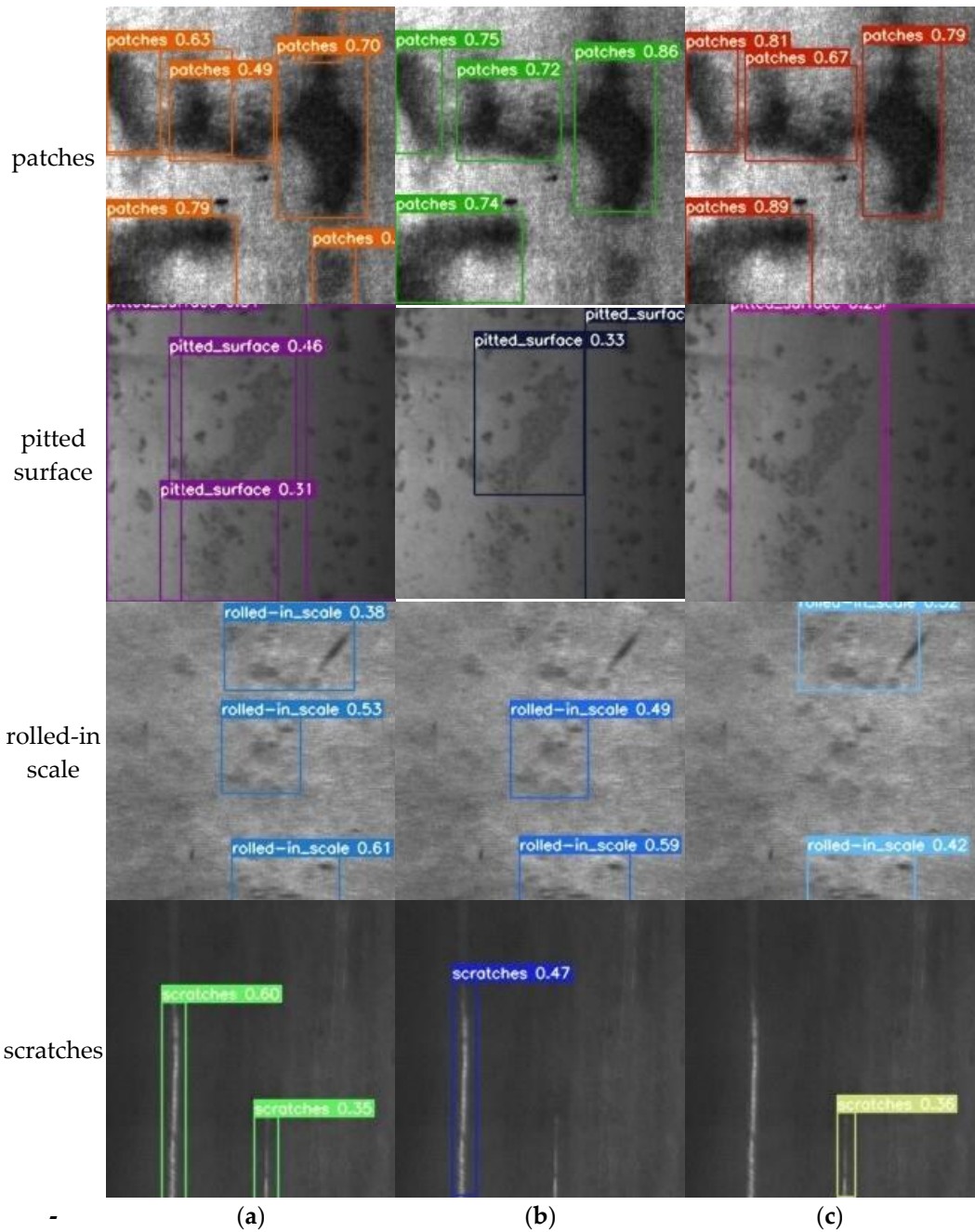

**Figure 9.** Comparison of detection results: (**a**) our algorithm; (**b**) YOLOv5 algorithm; (**c**) original YOLOv7 algorithm.

In order to more closely verify the superiority of the improved YOLOv7 algorithm compared with other algorithms, a comparison experiment was performed with five other classical algorithms that have been proposed. The experimental results are shown in Table 2. Compared with the other algorithms, the AP values of the proposed algorithm are not as good as the other algorithms for inclusion defects. However, the AP values and mAP values for other typical defects are higher than the other algorithms, and the mAP value is 6.6% higher than the YOLOv7. In summary, the proposed algorithm has a high practicality for the detection of strip defects in industrial production.

**Table 2.** Comparison results with other algorithms.

| | mAP% | AP% | | | | | |
|---|---|---|---|---|---|---|---|
| | | Crazing | Inclusion | Patches | Pitted Surface | Rolled-In Scale | Scratches |
| YOLOv5 | 77.80 | 40.60 | 81.00 | 96.70 | 98.20 | 70.10 | 80.30 |
| YOLOv7 | 76.30 | 48.10 | 76.50 | 94.80 | 99.50 | 67.00 | 72.00 |
| YOLOX | 73.37 | 46.06 | 73.26 | 86.58 | 83.55 | 52.80 | 97.98 |
| SSD | 75.43 | 62.72 | 75.63 | 94.31 | 71.46 | 65.89 | 82.54 |
| RetinaNet | 67.56 | 45.65 | 68.34 | 89.99 | 81.52 | 58.60 | 61.27 |
| Ours | 82.90 | 68.90 | 68.30 | 97.80 | 99.50 | 73.30 | 89.30 |

Note: AP = Average precision; mAP = Mean AP; SSD = Single-shot multi-box detection; YOLO = You Only Look Once.

## 5. Conclusions

Strip steel often inevitably produces some defects due to the level of production technology and the impact of external factors. However, in the image of strip surface defects, the shapes of similar defects are different, and there are similarities between different defects. Undoubtedly, it brings great obstacles to the defect detection algorithm. The conventional target detection algorithm can not well meet the requirements of the actual industrial production process of strip defect detection accuracy rate and inference speed. In this study, an improved YOLOv7-based detection algorithm to meet the requirements of accuracy and inference speed for strip steel defect detection is proposed. The value of this algorithm lies in adding the ConNeXt module to the backbone of YOLOv7, replacing the ELAN structure of the original model with the improved C3C2 structure in the head of YOLOv7, and embedding the CBAM attention module in the original pooling module. Among the six typical defects in the NEU-DET dataset, the proposed algorithm improves by 6.6% in mAP compared to the original model YOLOv7. Compared to the other single-stage target detection algorithms, SSD and RetinaNet, the mAP values increased by 7.47% and 15.34%, respectively. Compared to the YOLOX algorithm and YOLOv5, the detection accuracy of the other five defects is improved, except for the detection accuracy of inclusion defects, which is slightly decreased. Of course, there are still some shortcomings in our proposed algorithm; for example, the defect detection effect for the crazing and inclusion is still weak. We intend to further explore how to improve the algorithm's ability to identify these two types of defects through experiments in the future.

**Author Contributions:** Methodology, R.W. and X.M.; Software, F.L. and R.W.; Formal analysis, H.C.; Investigation, X.M. and X.Y.; Data curation, L.C. and Z.P.; Writing—original draft, F.L. and R.W.; Writing—review & editing, R.W. and X.M.; Funding acquisition, R.W. and X.M. All authors have read and agreed to the published version of the manuscript.

**Funding:** This study was co-supported by the industry-university-research innovation fund projects of China University in 2021(No. 2021ITA10018); the fund project of the Key Laboratory of AI and Information Processing (No. 2022GXZDSY101); the Natural Science Foundation Project of Guangxi, China (No. 2018GXNSFAA050026); the Key R&D Program Project of Guangxi, China (No. 2021AB38023); the basic ability improvement project for young and middle-aged teachers of universities in Guangxi, China (No. 2022KY0058).

**Institutional Review Board Statement:** Not applicable.

**Informed Consent Statement:** Not applicable.

**Data Availability Statement:** Data is contained within the article.

**Conflicts of Interest:** The authors declare no conflict of interest.

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
