# Peer review of "Development of an Improved YOLOv7-Based Model for Detecting Defects on Strip Steel Surfaces"

_coatings, doi:10.3390/coatings13030536_

Round 1

Reviewer 1 Report

In the manuscript entitled “Development of an improved YOLOv7-based model for detecting defects on strip steel surface”, R. Wang et al. have improved YOLOv7-based model to detect strip steel surface defects.

In the introduction the authors should compare the common methods/technology used to detects on strip steel surface with the propose methodology, stressing on the advantages and disadvantages.

Moreover, the authors should indicate the ISO Standard procedure used to detect those types of defect; which is the accuracy of their proposed technology? The authors should compare the results obtained by their proposed method with those obtained with the standard/common methodology.

How were the images reported in Figure 1 and 9 obtained? Discuss, and indicate also the scale bar.

In general, the arguments reported in manuscript expect a deep readers’ knowledge on the YOLOv7 model; in my opinion, the authors should turn to a wider readers with no specific knowledge on the YOLOv7 and other network structures. Therefore, they should explain in more easier and comprehensive way the proposed methodology, considering that the readers of coating journal should have no previous knowledge on the network structure, etc. For this observation, I suggest to submit the manuscript to another more specific journal.

Author Response

Dear Editors and Reviewers:

Thanks for your letter and for reviewer’s comments concern our manuscript. Those comments are valuable and helpful for revising and improving our paper, as well as the important guiding significance to our researches. We have studied all comments carefully and have made correction which we hope meet with approval. Revised portion are marked with underline in the paper.

Thank you and best regards.

Yours sincerely,
Rijun WANG

Reviewer 2 Report

1.       Summary

Presented article introduces surface defects detection model based on YOLOv7 neural network with respect to overall real-time image processing and classification into classes defined by the need of industry. Based on introduced requirements, the new, improved architecture of YOLOv7 model is presented. The contribution of article is in three areas: 1) improved YOLOv7 model architecture for defects detection on industrial strip steel, 2) YOLOv7 backbone modification for better features extraction with focus on the network inference improvement, 3) YOLOv7 head modification for the improvement of attention mechanism. The article provides good overview of all steps and explains methodology and experimental evaluation.

2.       Citations and resources

Cited references are relevant, some references for generally known models and methods are missing and should be provided based on my detailed recommendations (YOLOv7, YOLOv5).

3.       Manuscript

The article has good structure and provides all necessary details for problem, methodology, and application understanding. Experimental design follows standards defined for similar types of research and provides results in clear structure.

3.1   General comments

Based on review criteria, I have following comments to submitted article.

(L: denotes the line number)

1.      The Introduction section and Related Work section are using very complex and complicated sentence constructions. Sentences should be simplified to achieve better readability and easier understanding. Some sentences exceed 5-6 lines of text and are very hard to understand for readers. Moreover, comma rules are not applied correctly – please apply Oxford comma rules correctly only for enumerated lists.

2.       As the first references to industrial problem, references [1,2,3] are provided. Are there any references describing the problem from e. g. material industry point of view to explain problem in detail?

3.       Please, don’t use underscore for the defect names (e. g. L:33), use the form pitched surface, rolled-in scale instead.

4.       All the abbreviations, even those specific for convolutional networks and YOLO architectures, should be explained by the first use. For example, mAP, C3, C3C2, ELAN, CBAM, IoU, and others.

5.       L:166-167 – previous work is mentioned. Is there any reference to this work?

6.       The section L:170-182 contains 2 sentences only. Could you, please, simplify the structure to improve the ease of read?

7.       L:186, L:187 – YOLOv7 and YOLOv5 architectures are mentioned. However, both models have different, official and unofficial versions. Could you clearly state, which version (including reference) is used in your proposed work?

3.2   Specific comments

L: 203-214 The section “Loss function and label assignment” (L:203-214) defines three parts of the function – coordinate loss, target confidence loss, and classification loss. The formula later contains loss variables Lcls, Lobj, and Lloc. Please, provide some mapping to these variables, e.g., “coordinate loss (Lloc), target confidence loss (Lobj), and classification loss (Lclass).

L: 229 Change the format of the line, please.

L:237 You mentioned 4 different versions of ConvNeXt structure (T/S/B/L). However, 5 versions are listed – T/S/B/L/XL namely. Please, consolidate the section.

L: 278 The Figure 6 is mentioned, but the Figure 6 is missing in the article!

L: 328 The term dissect has strong anatomical meaning. Could you replace it by the term analyze (or similar in meaning), please?

L: 334 Reference [29] format.

L: 339-340 I don’t understand, how exactly is the NEU-DET dataset divided/used for training/validation/testing. If I understand it well, there is 90% data used for training and 10% for testing and then for the training subset 10% is used for validation. Is this right? If so, rework this to lines to state clearly the ratio for data split, please.

L: 372 You are mentioning the model size here (in some memory size I guess). It could be helpful to have this parameter (or the number of parameters) for each of the model. This is up to your decision, whether you will add it or not.

L: 386-390 Please, be aware of the smaller page size and zoom (in width) presented P-R curve figures, please.

L: 399-406 Is it possible to add the text with the defect name for each row of pictures, please?

4.       Reproducibility

The dataset used for the model preparation and experimental work is publicly available.

Author Response

(The authors gave the same response as above.)

Reviewer 3 Report

Journal Name: Coatings

Manuscript ID: coatings-2214876

Manuscript Title: Development of an improved YOLOv7-based model for detecting defects on strip steel surface

Comments to Authors and Editors

            This study focuses on the Development of an improved YOLOv7-based model for detecting defects on strip steel surface, which is an interesting topic.

            This paper carefully produced with a scientific quality is good. The appropriateness of the research techniques and a good understanding of the research methodology was used. The work was well planned and executed however there is a lack of results presentations. The kinds of literature reviewed in this manuscript and in-depth knowledge of the field are to be required.

            The following clarification must be made to improvise the readability of the manuscript.

1. The abstract of the article is not clear and concise.

2. In the introduction section add recent literature published.

3. Mention the novelty of the adopted research in the introduction section.

4. English of the manuscript must be polished throughout the manuscript. Carefully correct the typographical mistakes in the entire manuscript.

5. The introduction and Related works section are like a thesis report, so completely revised as per the journal requirement.

6. The study materials section will be completely revised per the journal requirement.

7. High-quality Figures to be provided for better readability with proper legend and labels.

8. To be provided with the detailed proposed model analysis in the results and discussion in the text.

9. The Methodology validation section is to be completely revised as per the journal requirement. Is this YOLOv7-based model, significant for your analysis? Justify with standard literature.

10. Results and discussion must be supported by standard literature.

11. To be provided with the detailed SEM morphology analysis in the images.

12. The conclusion is needed to write more precisely with the application of this existing methodology.

The article may be acceptable for publication after clarifying the minor revision.

Author Response

(The authors gave the same response as above.)

Round 2

Reviewer 1 Report

The revised version of the manuscript is acceptable for publication.